# Complementary Dual Approach for In Silico Target Identification of Potential Pharmaceutical Compounds in Cystic Fibrosis

**DOI:** 10.3390/ijms232012351

**Published:** 2022-10-15

**Authors:** Liza Vinhoven, Frauke Stanke, Sylvia Hafkemeyer, Manuel Manfred Nietert

**Affiliations:** 1Department of Medical Bioinformatics, University Medical Center Göttingen, Goldschmidtstraße 1, 37077 Göttingen, Germany; 2Clinic for Pediatric Pneumology, Allergology and Neonatology, Hannover Medical School, Carl-Neuberg-Strasse 1, 30625 Hannover, Germany; 3Biomedical Research in Endstage and Obstructive Lung Disease Hannover (BREATH), German Center for Lung Research, Carl-Neuberg-Strasse 1, 30625 Hannover, Germany; 4Mukoviszidose Institut, In den Dauen 6, 53117 Bonn, Germany; 5CIDAS Campus Institute Data Science, Goldschmidtstraße 1, 37077 Göttingen, Germany

**Keywords:** cystic fibrosis, docking, ligand-based drug design, target-based drug design/target identification, virtual screening

## Abstract

Cystic fibrosis is a genetic disease caused by mutation of the CFTR gene, which encodes a chloride and bicarbonate transporter in epithelial cells. Due to the vast range of geno- and phenotypes, it is difficult to find causative treatments; however, small-molecule therapeutics have been clinically approved in the last decade. Still, the search for novel therapeutics is ongoing, and thousands of compounds are being tested in different assays, often leaving their mechanism of action unknown. Here, we bring together a CFTR-specific compound database (CandActCFTR) and systems biology model (CFTR Lifecycle Map) to identify the targets of the most promising compounds. We use a dual inverse screening approach, where we employ target- and ligand-based methods to suggest targets of 309 active compounds in the database amongst 90 protein targets from the systems biology model. Overall, we identified 1038 potential target–compound pairings and were able to suggest targets for all 309 active compounds in the database.

## 1. Introduction

Cystic fibrosis (CF) is one of the most common genetic diseases prevalent among the population of Caucasian ancestry, where it affects approximately 1 in 3000 newborns [1,2,3]. It is caused by mutations of the cystic fibrosis transmembrane conductance regulator (CFTR) gene [4], which encodes a membrane protein that serves as a chloride and bicarbonate channel in the exocrine epithelia of various organs, thereby regulating the viscosity of the mucus lining [5]. Defective CFTR, therefore, has severe implications throughout the body, its major hallmarks being recurrent pulmonary infections and pancreatic insufficiency [4,5]. Of the currently known 2100 mutations of the CFTR gene, several hundred have been shown to be disease causing [6,7,8]. These mutations cause disturbances throughout CFTR’s intricate and delicately balanced biogenesis, which, even in its wild-type (wt) form, only converts 20–40% of the transcripts into a fully functional protein [9]. In order to make handling and working with the multitude of mutations easier, they are categorized into different classes, depending on what kind of defect they cause. Originally, four major mutation classes were proposed, and over the years, this has been expanded to seven classes (I: no CFTR synthesis; II: CFTR trafficking defect; III: CFTR dysregulation; IV: defective gating; V: reduced CFTR transcription; VI: less stable protein; and VII: no CFTR mRNA) [10,11,12,13,14]; however, as many mutations cause multiple defects, an expanded, combinatorial classification system was proposed [15]. Here, mutations are assigned to groups of all possible combinations of the single-defect mutation classes, resulting in a more comprehensive grouping. In addition to the multi-defect mutations, patients often carry more than one mutation. These factors lead to a vast range of geno- and phenotypes, which makes the development of effective causative therapeutics especially challenging. In recent years, different small-molecule therapeutics have been developed for clinical applications, which improve the CFTR function by directly targeting the CFTR protein and not just alleviating symptoms of CF-patients. Currently, four pharmaceutical drugs, different combinations of four compounds, are approved and available as causative therapy to some CF patients [16,17,18,19]. The active compound in the first approved drug, Kalydeco, is Ivacaftor, a CFTR potentiator which is approved mainly for gating mutations [20,21]. The other drugs are combination therapies [18,19], i.e., they contain two or more active compounds and thereby target multiple defects. Orkambi contains Ivacaftor and Lumcaftor, a CFTR corrector, which acts as small-molecule chaperone to correct the folding defect of some class II mutations [22,23], and Symdeco, which contains Ivacaftor and Tezacaftor, an alternative CFTR corrector [24]. The most recent drug, Kaftrio (known as Trikafta in the US), is a triple combination of Ivacaftor, Tezacaftor and Elaxacaftor, which also acts as CFTR corrector [25,26]. Still, for about 10% of patients, especially those with rare mutations, there is no causative medication available [27]. In an effort to find effective treatments for all patients, the search for CFTR modulators, especially synergistic compound combinations, is ongoing [28,29]. Thousands of compounds are being tested in different cell-assays, often in high-throughput screens, leading to large amounts of data and various candidate substances [20,30,31,32,33,34,35,36,37,38,39,40,41]. In order to structure and collect the compounds tested as CFTR modulators, we previously developed the publicly available database CandActCFTR (candactcftr.ams.med.uni-goettingen.de), where compounds are annotated and categorized according to various characteristics, including their mode of action and order of interaction with CFTR [42,43]. When analysing the data, it becomes apparent that for about 70% of the active compounds, it is unknown whether they affect CFTR directly through physical interaction, or indirectly through its interactome. To support the elucidation of their mode of action, we previously developed the CFTR Lifecycle Map (cf-map.uni-goettingen.de), where we used a systems biology approach to create a human- and machine-readable model of the CFTR maturation pathway in cells [44,45]. The CFTR Lifecycle Map is written in the standardized SBGN Process Description format and comprises detailed representations of the molecular interactions and pathways CFTR undergoes during its entire biogenesis. It contains 156 reactions with 262 different molecular entities, including 170 biomacromolecules, mainly proteins. Here, we now connect the two resources in order to shed light on the mechanism of action of active compounds by identifying their targets. For this purpose, we are using and combining two different reverse screening approaches. Traditionally, in drug discovery, virtual screening approaches are applied to find bioactive compounds that bind to a specific target protein. In reverse screening, the opposite approach is employed to identify the target proteins of active compounds [46,47]. This is becoming increasingly important in order to predict drug side effects and for drug repositioning, where existing drugs are repurposed for other disorders. Several computational methods exist for reverse screening, which, similarly to traditional virtual screening, can generally be divided into three categories [47]. The first class is target-based approaches, mainly docking, which requires high-quality protein structures and has high computational costs. The two other classes are ligand based, either on their shape or their pharmacological features. These require a solid data foundation of known ligand–target interactions as reference databases. In the last decade, reverse screening approaches have been applied to a range of use cases, especially to find targets of natural compounds [46,47]. For example, a molecular shape-based method was employed to identify the cyclin dependent kinase (CDK2) as target of the phytochemical curcumin as possible explanation for its cancer-preventive properties [48]. Furthermore, inverse docking, i.e., a target-based approach, has been used to, amongst others, study different helicases in Zika viruses as targets of ligands from a flowering plant [49], investigate the effect of thyme derived thymol on fat deposition [50], and shed light on the antitumor targets of a library of natural bioactive compounds [51]. While most studies use either one or the other approach, we here use both a target- and ligand-based approach independently and combined to identify possible targets in the CFTR Lifecycle Map of the active compounds from CandActCFTR. By using the two approaches, we were able to include more potential protein targets than by exclusively using docking or shape-based approaches. Ultimately, we tested 309 active compounds against 90 targets, resulting in almost 30,000 possible combinations. Of these, 1038 unique target–compound pairings were identified with graduated confidence levels in the range of 1–5. Importantly, we could suggest at least one target for each active compound in the database, thereby bringing the elucidation of their mechanism of action one step closer and serving as a basis for finding novel compounds (classes) and predicting synergistic compound combinations. This application shows how systems medicine disease maps, such as the CFTR Lifecycle Map, can be utilized for in silico target identification and to provide the means to fill knowledge gaps and support drug design.

## 2. Results

### 2.1. PDB Targets

In order to suggest possible mechanisms of actions of active compounds in the CandActCFTR database [43], possible protein targets that are involved in CFTR biogenesis were selected from the CFTR Lifecycle Map [45]. For this purpose, the Protein Data Bank [52] was queried for experimental X-ray or CryoEM structures of all proteins within the CFTR Lifecycle Map. The list was then filtered according to different criteria such as structure completeness, resolution and experimental method. This list was narrowed down to 35 PDB structures, including that of wt-CFTR, which are listed in Appendix A, together with their experimental specifications and their role in the CFTR biogenesis. Overall, for 35 of the 170 proteins present in the CFTR Lifecycle Map, appropriate PDB structures could be found, which cover a diverse range of functions and stages in the CFTR biogenesis. This will be especially important to develop combination therapies with synergistic effects that influence CFTR at different stages of its lifecycle. However, for some steps in the CFTR lifecycle, specifically at the transcription stage, no docking-appropriate PDB structures of CFTR interactors could be found. To remedy this situation, structure predictions from the AlphaFold database (https://alphafold.ebi.ac.uk/, accessed on 9 September 2022) [53,54,55] were considered as alternatives. Unfortunately, when comparing the results of the blind cross-docking between a set of reference protein structures with co-crystallized ligands and their predicted counterparts, the predicted structures resulted in much less accurate docking results. Since the aim of this study was to find potential targets for our query ligands, rather than of putative ligands for specific proteins, the inaccurate docking results from predicted structures could potentially bias the overall results. It was therefore decided not to use the predicted structures from the AlphaFold database and continue with the subset of 35 experimental structures from the PDB database. Nonetheless, except for the transcription stage, the 35 targets are well distributed across the CFTR Lifecycle Map. More precisely, in addition to CFTR itself, 9 potential targets are involved in translation, folding and ER quality control, 7 are associated with the secretory pathway, 5 with endocytosis, and 13 are involved in CFTR activity.

### 2.2. Docking

All active compounds from the CandActCFTR [42,43] database were docked against all 35 targets using the Smina [56] and QuickVina-W (qvina-W) [57] docking programs. After docking, two methods (2.1) were used for post-processing. The first method [51] (method I) filters out false positives and ultimately results in a list of the overall best-scoring target–ligand pairings, while the second method [58] (method II) also filters out false positives but then calculates the most likely target for each ligand. A comparison of the results from all four approaches can be seen in Figure 1A. Method I resulted in a list of 21 target–ligand pairings for the docking results from the qvina-w docking program. Of the 48 high-scoring pairings, 19 were also among the ones identified by method II.

Using the results from the Smina docking program, 45 target–ligand pairings were identified using the method I, 30 of which were also identified by the second approach. 

When comparing the results from the two docking programs, method II identified 43 common target–ligand pairings, while there were no common pairings found by both method I approaches.

Figure 1 shows the number of ligands attributed to the targets by all methods combined. On average, the mean Tanimoto similarity amongst ligands associated with the same target was 0.25 on average, indicating that there is no bias towards a specific compound class for one target. As can be seen, PRKACA (the catalytic subunit α of protein kinase A) (108 ligands) and CSNK2A1 (Casein kinase II subunit α) (60 ligands) have a lot of compounds ascribed to them by at least one approach. As both analysis methods take into account average scores for each ligand over all targets, the docking results for PRKACA and CSNK2A1 were removed from the data, and the entire analysis of the docking results was repeated, in order to eliminate bias from these two targets.

Figure 2B shows that the number of ligands identified per target are much more evenly distributed after removing the PRKACA and CSNK2A1 from the docking data. The target with the most ligands associated with it here is RAB5A (Ras-related protein Rab-5A) (47 ligands). On average, the mean Tanimoto similarity amongst ligands associated with the same target was 0.21, so they appear to be structurally different.

The number of targets per ligand of all pairings identified by method I (including and excluding the results with PRKACA and CSNK2A1) can be seen in Figure 3. As can be seen, the number of potential targets for each ligand is relatively low, indicating specific interactions. Most ligands are only associated with one or two targets, one is associated with three and four targets, respectively, two to five targets, and only one ligand is associated with six targets. The ligand with the most targets associated (ligand 2144) is the deoxyribose dATP, which is predicted to bind to the Ras-related proteins RAB4A, RAB5A, RAB7A, RAB9A, and RAB11B and the secretion-associated Ras-related GTPase 1A (SAR1A), all of which are GTP-binding proteins. 

Figure 3B shows the Protein–Ligand-Interaction network for all pairings identified by method I (including and excluding the results with PRKACA and CSNK2A1). Each edge between two targets stands for a compound found for both of them, the width of the edge representing the number of compounds and the size of the node (targets) representing its degree, i.e., how many other nodes it is connected to. Most targets have only one or two potential compounds in common, except PRKACA and CSNK2A1, which share four compounds.

As can be seen in the Venn diagram in Figure 2A, the consensus amongst the different approaches was similar compared to when all targets were used. For the qvina-w docking program, method I led to ten target–ligand parings, six of which were also identified by method II. The results from the Smina docking program led to 39 pairings using method I, 25 of which were also found with method II. Comparing the two docking programs, method II identified 32 common pairings, and method I identified only one common pairing. Interestingly, the ligand in this pairing is Lumacaftor (InChI Key: UFSKUSARDNFIRC-UHFFFAOYSA-N), also known as VX-809, a well-known and clinically approved CF-drug, known to bind to CFTR directly. Here, however, it is predicted to bind to the Ras-related protein Rab-7a (RAB7A), a protein involved in endocytosis.

Two different binding pockets and binding poses were identified by both docking programs, Smina and qvina-w, in accordance. The binding pockets can be seen in Figure 4, where the poses predicted by Smina are coloured in blue, and the ones predicted by qvina-w are green. Binding pocket A corresponds to the GTP binding site of RAB7A. The two docking poses calculated by Smina and qvina-w, respectively, have an RMSD of 2.30 Å in pocket A and 2.24 Å in pocket B. Figure 5 shows the 2D poseview of Lumacaftor docked into both pockets by both docking programs. The black dotted lines represent hydrogen bonds, the green spline sections represent hydrophobic interactions and the green dotted lines show π-π stacking or π–cation interactions. As can be seen in Figure 5, in pocket A, for both predicted binding poses, Lumacaftor interacts with various residues via hydrogen bonds, and it undergoes hydrophobic interactions with tyrosine-37 and an π–cation interaction with lysine-126. However, while the binding pose calculated by Smina shows a π–cation interaction with the Magnesium ion in the GTP binding site, the binding pose predicted by qvina-w suggests a π-π stacking interaction with tyrosine-37. In binding pocket B, both binding poses suggest Lumacaftor undergoes hydrogen bonding with serine-72. The binding pose predicted by qvina-w suggests an additional hydrogen bond with glutamine-71, while the one predicted by Smina shows π-π stacking with tryptophane-102, as well as hydrophobic interactions. Overall, both docking programs predict a higher binding affinity for pocket A (Smina 13.4 kcal/mol, qvina-w 12.4 kcal/mol) than for pocket B (both 10.8 kcal/mol).

Two other target–ligand pairings were identified by three of the four approaches after the exclusion of PRKACA and CSNK2A1 from the docking data. The Ras-related protein Rab-4a (RAB4A) was predicted to interact with ligand2995 (InChI Key: PSPRNONTFBJUDQ-SCFUHWHPSA-N), an ATP analogue. Figure 6A shows the ligand at the GTP binding site of RAB4A. There it is suggested to be coordinated via hydrogen bonding with leucine-155, serine-158, alanine-154 and lysine-124, as well as a π-π stacking interaction with phenylalanine-35. For this pairing, qvina-w calculated a binding affinity of −10 kcal/mol, and Smina calculated a binding affinity of −12.7 kcal/mol.

Furthermore, Derlin-1 (DERL1), also known as degradation in endoplasmic reticulum protein 1, was predicted to interact with Equol (ligand104, InChIKey: ADFCQWZHKCXPAJ-GFCCVEGCSA-N). Figure 6B shows the predicted binding pocket of Equol in DERL1, where it is predicted to undergo π-π stacking interactions with tryptophane-106 and 18. Here, qvina-w calculated a binding affinity of −8.9 kcal/mol, while Smina calculated a binding affinity of −11.1 kcal/mol.

### 2.3. Ligand Similarity Approach

For the ligand-based similarity approach, known ligands for the 35 selected targets were collected from the databases BindingDB [59] (https://www.bindingdb.org, accessed on 9 September 2022) and ChEMBL (https://www.ebi.ac.uk/chembl/, accessed on 9 September 2022) [60,61,62]. Overall, 6787 target–ligand interactions could be found in the BindingDB and 8536 were collected from ChEMBL. When merging the two datasets, 3631 of target–ligand interactions could be found in both databases, resulting in a combined dataset of 11,692 unique interactions. From ChEMBL, ligands could be found for 22 targets, of which 15 also had ligands listed in the BindingDB. The targets and the number of ligands per target from each database are listed in Appendix A. The most ligands could be found for the chaperone HSP90AA1 (3112 ligands), followed by the protein kinases PRKACA (1730 ligands), CSNK2A1 (1710 ligands) and PRKAA2 (1551 ligands), which are well-known pharmacological targets. For CFTR, 1012 ligands could be found, while less than 1000 compounds were found for the remaining targets. Hence, ligand-based similarity comparisons were conducted for 22 of the 35 targets.

For this purpose, the molecular fingerprints of the reference compounds and the query compounds were calculated and compared pairwise to each other by calculating the Tanimoto similarity. Using a similarity cut-off of 0.75, similar compounds between reference and query ligands could be found for eight targets (Appendix A). Figure 7A shows that the most compounds were found for CFTR (52 compounds), followed by the catalytic subunit of the Serine/threonine-protein phosphatase 2B (PPP3CA; 34 compounds), and the kinases PRKACA (11 compounds), PRKAA2 (11 compounds) and CSNK2A1 (10 compounds). Two compounds could be found for each of the chaperones HSPB1 (also known as Hsp27) and HSP90AA1, and one for ATPase 2 (ATP2A1). Of the 52 compounds identified for CFTR, 13 were previously known to interact with CFTR directly. Reversely, not all compounds reported as directly interacting with CFTR in the CandActCFTR database were also identified via ligand similarity, showing that the reference ligands from ChEMBL and the BindingDB do not exhaustively include all known ligands.

Figure 7B shows the Protein–Ligand-Interaction network of the results. As can be seen, the kinases PRKACA, PRKAA2 and CSNK2A1 share the most compounds with each other. All 11 compounds associated with them are shared between PRKACA and PRKAA2, 9 of which they also share with CSNK2A1. All three kinases share the same seven compounds with CFTR and the same two with HSPB1. PRKACA and PRKAA2 additionally share two identical compounds with PPP3CA, which also shares one compound with each CFTR and HSP90AA1, who share another compound amongst them.

As the ligand-based similarity approach does not depend on protein structures, the dataset was extended by including all remaining potential targets from the CFTR Lifecycle Map, resulting in a list of 168 proteins. The same procedure used for the small dataset was used here to identify potential targets of active compounds in the CandActCFTR database. For all targets combined, 36,851 target–ligand interactions could be found in the BindingDB, and 29,020 were found in ChEMBL. There was an overlap of 15,984 target–ligand interactions, resulting in a combined dataset of 49,887 unique interactions. Ligands for 75 targets were collected from ChEMBL and ligands for 54 targets were collected from the BindingDB, with an overlap of 52 targets between them. More than 5000 ligands could be found for the histone deacetylase 6 (HDAC6, 6815 ligands), the phosphodiesterase 4D (PDE4D, 6769 ligands), the glucocorticoid receptor NR3C1 (5063 ligands) and the adenosine A2B receptor (ADORA2B, 5032 ligands). Again, the high number of ligands present in the database indicates that they are pharmacological interesting targets. For 8 different targets, more than 1000 ligands could be collected, and less than 100 ligands were found for 46 targets.

The ligand similarity comparisons resulted in potential targets for 108 compounds, distributed across 25 targets. Figure 8A shows the number of compounds identified per target. Again, most compounds were found for CFTR (52 compounds), due to the general bias towards CFTR-relevant compounds amongst the query ligands from CandActCFTR. More than 40 compounds were linked to NR3C1 (42 compounds), the beta-2 adrenergic receptor (ADRB2, 40 compounds), followed again by the catalytic subunit of the Serine/threonine-protein phosphatase 2B (PPP3CA; 34 compounds). For the remainder of the targets, less than 15 compounds were found to be similar to the known ligands. 

Figure 8B shows the Protein–Ligand-Interaction network of the results. As can be seen, the targets with the most shared compounds are NR3C1, ADRB2 and PPP3CA. PPP3CA shares all of its 34 compounds with ADRB2 and NR3C1. The next targets with the most shared compounds are again PRKACA, PRKAA2 and CSNK2A1, which, as already described above, have 11 and 9 shared compounds, respectively. Additionally, CFTR and NR3C1 share an overlap on 9 compounds. Overall, these seven targets (ADRB2, CFTR, CSNK2A1, NR3C1, PPP3CA, PRKAA2 and PRKACA) are the most connected between each other. The remaining targets share a smaller number of compounds with any of the other targets.

### 2.4. Combined Approach

In order to obtain the most comprehensive overview of potential target–compound interactions, the results from both the structure-based docking and the ligand-based approach were combined.

Overall, a total of 1038 unique target–compound pairings were found, 757 via the structure-based approach and 290 via the ligand-based approach (Appendix A). The pairings were assigned confidence scores in the range of 1–5, depending on the number of approaches they were identified by. Hence, a score of 5 is assigned when pairings are identified by all methods, i.e., all four target-based approaches as well as the ligand-based approach. The score therefore represents the consensus amongst the different approaches used. The highest score reached was 4, by the RAB7A–Lumacaftor pairing. Overall, 8 pairings have a score of 3, 117 pairings have a score of 2, and the remaining 912 pairings were identified by only 1 approach. No significant structural similarities could be found between the compounds with higher confidence levels, indicating that there is no bias present amongst them.

Furthermore, nine target–ligand pairings were identified by both the target- and ligand-based approach. Of the nine pairings, five ligands were associated with CFTR and two to each of PRKACA and PRKAA2. Four of the five compounds *(Corr4a, InChIKey RDOBOPJBMQURAT-UHFFFAOYSA-N; CHEMBL4471507, InChIKey DBGTUHVUTOSJOJ-UHFFFAOYSA-N; CHEMBL4574818, InChIKey KHNUPLUQJFLSLN-UHFFFAOYSA-N; CHEMBL4435663, InChIKey XTWGHYUHSFTZLL-UHFFFAOYSA-N)* associated with CFTR are similar in structure, but the fifth (InChIKey NOGRVEQYQUCZIW-UHFFFAOYSA-N) differs substantially. One compound, apigenin, was associated with both PRKACA and PRKAA2. Apart from apigenin, PRKAA2 was associated with Kaempferol (InChIKey IYRMWMYZSQPJKC-UHFFFAOYSA-N) and PRKACA was associated with Biochanin (InChIKey WUADCCWRTIWANL-UHFFFAOYSA-N).

Potential targets were found for all 309 active compounds from the CandActCFTR database (Appendix A). The distribution of compounds across the targets is visualized in Figure 9. Targets which were associated with at least one compound are displayed in colour, the colour itself representing how many compounds they were associated to, ranging from 1 compound (yellow) to 120 compounds (red). The target with by far the most compounds associated with it was PRKACA (120 compounds), followed by CFTR with 76 compounds, CSNK2A1 with 72 compounds and RAB5A with 52 compounds. Of the remaining targets, 27 had between 10 and 50 ligands associated with them, and 20 targets had less than 10 compounds. When looking at the binding sites of the potential CFTR ligands identified by the target-based approach, mostly five main binding sites were identified (Figure 10). Two of these correspond to the ATP-binding sites in the nucleotide binding domains 1 and 2. One was close to the experimentally resolved binding site of Lumacaftor (coloured in blue), one was close to the one of Ivacaftor (coloured in purple) [63], and there was one additional binding site in the transmembrane domain 2. 

Conversely, for the majority of compounds, 2–4 targets were identified, while a single target was identified for only 15 compounds. Between 5 and 10 targets were identified for 49 compounds, 1 compound has 11 targets associated with it and 6 compounds have 12 targets associated with them. Interestingly, all of these 6 compounds are almost identical in structure and vary only in stereochemistry or side chain. These compounds were all predicted to target the kinases CSNK2A1, PRKAA2, PRKACA, PRKCE and WNK1, but also other targets, namely, ADOR2B, ADRB2, CFTR, NR3C1 and TNF.

## 3. Discussion

When searching for new drugs experimentally, especially in high-throughput screens, where thousands of compounds are tested simultaneously, the mechanism of action of promising compounds often remains unclear. Knowing the mechanism of action, however, is helpful and important for a number of reasons. It is useful in order to identify novel drug targets, find new classes of potentially active compounds, and to aid lead optimization. Furthermore, it can be helpful for the early identification of potential side effects. Elucidating the mechanism of action of promising compounds is especially important in the drug discovery for cystic fibrosis, as it is crucial for the development of combination therapies. Here, we bring together two previously developed resources of community-derived knowledge on cystic fibrosis, namely, the CandActCFTR database and the CFTR Lifecycle Map, to aid in the elucidation of modes of action for known active compounds. For this purpose, we use two different approaches to in silico target identification, a target-based approach and a ligand-based approach. 

Both approaches have different advantages and disadvantages when it comes to prerequisites. The ligand-based approach is much faster and less computationally expensive. However, in order to be most effective, it requires comprehensive ligand libraries of the targets in question. The target-based approach, on the other hand, requires complete and high-resolution structures of the target proteins. From the 170 potential targets in the CFTR Lifecycle Map, quality-sufficient protein structures could be found for 35 targets, and ligand collections could be found for 77 targets, with an overlap of 22 targets. By combining the approaches, it was thus possible to cover more than 50% of the potential targets in the CFTR Lifecycle Map. 

For the target-based approach, all active compounds from the CandActCFTR database were docked blindly into the 35 target structures, meaning that no prior binding site was defined, but the whole protein was searched. In order to produce comprehensive results, two different docking programs, QuickVina-W (qvina-w) and Smina, were used, which differ in their method to calculate binding affinities. Due to their different scoring functions, qvina-w and Smina produced different results, but with overall similar trends. Generally, the binding affinities calculated by Smina were slightly higher than those calculated by qvina-w. This is most likely caused by their different approaches to estimating ligand–receptor-based affinity. While qvina-w uses the empirical Vina scoring function [57], which is based on machine-learning [64], Smina uses a more physics-based approach [65]. However, the exact binding affinities are of secondary importance in this case, as the aim in target identification is not to calculate the most precise scores, but rather compare the likeliness of different matches to find potential target–ligand pairings. Therefore, in order to normalize the results and remove false positives, two post-processing methods were applied to the docking data independently. Method I results in a list of high-ranking target–ligand pairings, while method II identifies the most likely target for each ligand. Overall, however, the two methods produce similar results. Due to the high number of ligands, method II results in a lot of more potential pairings; however, a high portion of pairings identified by method I are also suggested by method II, underpinning both their validity, while still remaining non-redundant. Interestingly, two proteins, PRKACA and CSNK2A1, were identified in pairings significantly more often than the other ones. Both proteins are protein kinases, which have been previously found to be promiscuous targets [66,67]. With respect to the high number of ligands associated with the PRKACA and CSNK2A1, the results from the ligand-based approach support the results from the target-based approach. Again, the two kinases were amongst those with the most compounds associated with them. Interestingly, a majority of these compounds could be associated with both proteins, and additionally the kinase PRKAA2, which suggests a common, possibly promiscuous, binding motif amongst these kinases. In order to remove bias from PRKACA and CSNK2A1 from the docking calculations, the analysis was repeated without the data for these two kinases.

Overall, the majority of the predicted interactions (506 of the 1038 predicted interactions) involve proteins that play a role in the activity and regulation of CFTR at the plasma membrane. One explanation for this are the readouts of the experimental assays used to identify the active compounds in their original publications. Most assays use the ion conductance directly as a readout, so compounds that affect the activity of CFTR directly at the membrane are readily detected by these methods. Of the remaining predicted interactions, 161 involve proteins that play a role in CFTR translation and folding, 120 involve proteins of endocytosis and 108 involve protein of the secretory pathway. At total of 76 interactions involve CFTR directly, and only 67 interactions involve transcriptions factors or other proteins that influence CFTR transcription.

Remarkably, one of the protein-ligand pairings with the highest consensus amongst the different approaches was for the compound Lumacaftor (VX-809) with RAB7A. Lumacaftor is a well-known, clinically approved drug for patients with F508del mutations, as it acts as a small-molecule chaperone to correct the folding defect of F508del-CFTR. It is known to bind directly to CFTR, with its exact binding site elucidated by CryoEM (PDBs 7SVD and 7SVR) [63]. Here, however, it was suggested to potentially also bind to the Ras-related protein RAB7A, a GTP-binding protein involved in endocytosis, including that of CFTR [68,69,70]. When looking close at the predicted binding mode, two different binding sites are suggested for Lumacaftor in the RAB7A protein by both docking programs. In both pockets, Lumacaftor is predicted to interact with the protein via different hydrogen bonds, hydrophobic interactions and π-interactions. When looking at the binding mode for Lumacaftor to CFTR as shown in the CryoEM structure (7SVD) [63], the compound is coordinated mainly by hydrophobic interactions and only one hydrogen bond (Figure 11). Hydrophobic interactions are rather weak intermolecular interactions, which is probably why the binding affinity for Lumacaftor was calculated to be higher for RAB7A than CFTR by the docking programs. However, when looking at the raw docking results for CFTR, Lumacaftor is nonetheless amongst the top 10 highest ranking active compounds calculated by both docking programs. Furthermore, Hou et al. previously showed how RAB7A inhibition increases apical CFTR stability [70]. Hence, while certainly intriguing and requiring further investigation before confirmation, the pairing of Lumacaftor with RAB7A, rather than with CFTR, therefore does not undermine the potential validity of the results, as the scores calculated for the CFTR–Lumacaftor (Figure 12) pairing are only slightly lower than the one for RAB7A–Lumacaftor.

Another high-consensus protein–ligand pairing from the target-based approach was for the compound Equol with DERL1. Equol is an isoflavandiol oestrogen, known to bind to oestrogen receptor β [71]. It was also shown to be on the one hand a potent activator of F508del-CFTR, but on the other hand, it affects its misprocessing [72]. DERL1 is an ER membrane protein mediating the degradation of misfolded proteins such as misfolded CFTR [73,74]. Thus, the effect Equol has on F508del-CFTR misprocessing might be mediated by its binding to DERL1. 

Of the 170 different proteins in the map, either protein structures or ligand data could be found for 90 of them, which amounts to a coverage of 53%. This means that 80 proteins (47% of proteins in the map) could not be considered as targets, due to the lack of available data. As is often the case in in silico approaches, this effort to elucidate the mechanism of action of candidate compounds was hampered by the lack of a comprehensive data corpus. Nonetheless, by combining the ligand- and the target-based approach, potential targets were suggested for all 309 active compounds in the CandActCFTR database, distributed across 53 of the potential targets from the CFTR Lifecycle map.

It has to be noted that not all proteins involved in the CFTR biogenesis are equally suitable as targets for CFTR modulators. For example, chaperones and other proteins that are important for CFTR folding, ER quality control, and trafficking play a quite general role in cells and are not specific to CFTR alone. Targeting them can therefore potentially lead to unwanted side effects. Therefore, when selecting compounds and targets for experimental testing, more research is needed on which proteins can be targeted and which should rather not be interfered with.

Of course, experimental validation of protein–ligand interactions is always necessary and preferred to docking- and ligand-similarity approaches. However, where experimental data is not available or its generation not feasible, these in silico approaches are a good way to fill knowledge gaps, suggest potential interactors, and thereby narrow down the candidates for testing in the wet lab. These methods should therefore not be viewed as replacement for wet-lab experiments, but as preceding step to support experimental work and, in the meantime, at least provide indications of possible modes of action.

This project is part of the CandActCFTR project and serves as an essential connecting piece between the chemistry centric CandActCFTR database, which collects compounds tested as CFTR modulators, and the cell biology centric CFTR Lifecycle map, which is a systems biology model of the CFTR biogenesis. By bringing the two parts together here, we not only use and repurpose data from both resources, but are able to generate new knowledge. The structured knowledge in systems medicine disease maps can therefore directly be applied to drug design approaches. By using this comprehensive approach, we can now suggest mechanisms of actions for active compounds, propose potential drug targets and predict possible additive effects of different substance combinations.

## 4. Materials and Methods

### 4.1. Reverse Docking

In order to identify targets for the active compounds of the CF-specific compound database CandActCFTR [42,43], 37 potentially relevant protein targets were used for reverse docking. The proteins were selected using the CFTR lifecycle map [45], a systems biology model of the CFTR lifecycle. For this purpose, a KNIME [75] workflow was used to search all PDB [52] entries for all proteins in the map and then filter them according different criteria such as structure completeness, resolution and experimental method. This list was narrowed down to 35 PDB structures, including that of wt-CFTR, belonging to targets evenly distributed across the CFTR lifecycle. All target structures were prepared for docking using AutoDockTools 1.5.7 [76]. 

The active compounds from the CandActCFTR database [43] were used as ligand library. Structures were obtained in SMILES notation and converted and prepared for docking using Open Babel [77,78]. 

Docking of all compounds against all ligands was performed using Virtual Flow [79] with two different docking programs. Specifically, it was carried out using the Smina docking program [56], with the Vinardo scoring function [65] to calculate binding scores between ligands and targets, and the QuickVina-W [57] program, with the AutoDock Vina scoring function [64]. Docking was carried out as blind docking, meaning that no specific binding pocket was defined for each protein, but the entire protein was searched in order to account for structures without known binding site and targets with several, potentially unknown, binding sites. The search space in blind docking is significantly higher than when using pre-defined binding sites, which leads to higher probabilities of not finding the optimal conformation. Nevertheless, to obtain comprehensive results, the exhaustiveness, i.e., the number of runs calculated per ligand and target was increased to 100.

All docking calculations were run on the Scientific Compute Cluster at the joint data centre of Max Planck Society for the Advancement of Science (MPG) and University of Göttingen with the SLURM Workload Manager [80].

Post-processing of the docking was performed using KNIME [75,81] and Pyrhon. In order to remove false positives and obtain more reliable consensus results, two different approaches were used to evaluate target–ligand pairings.

The first approach proposed by Lauro et al. [51] normalizes the binding energies of each target–ligand pairing in a way that eliminates systematic errors and exclude false positives. To do so, the docking results are written in a matrix format and equation (1) was applied, where V is the new value, V_0_ is the binding energy from the docking calculate, M_L_ is the average binding energy of each ligand, and M_T_ is the average binding energy of each target.
(1)$$V=\frac{V_{0}}{\left[M_{L}+M_{T}\right]/2}$$

Promising pairings were then selected by setting a lower threshold at V = M + 3σ, where M is the average of the whole matric, and σ is its standard deviation.

The second approach by Kim et al. [58] uses a 2-directional Z-transformation on the docking-score matrix. Here, a target- and ligand-specific Z-score is calculated using Equations (2) and (3), where is $x_{i}$ the docking score specific to the query ligand i, ${\bar{x}}_{t}$ and ${\bar{x}}_{l}$ are the average scores of all targets and ligands, respectively, and $SD_{T}$ and $SD_{L}$ are the respective standard deviations.
(2)$$Z_{T}=\frac{\left(x_{i}-{\bar{x}}_{T}\right)}{SD_{T}}$$
(3)$$Z_{L}=\frac{\left(x_{i}-{\bar{x}}_{L}\right)}{{\mathrm{SD}}_{L}}$$

The combined $\;Z_{comb}$ is then calculated through$\;\;Z_{comb}=0.7*Z_{T}+0.3*Z_{L}$. In order to select the most likely target, the receptor with the lowest $\;Z_{comb}$ is selected for each ligand. The 2-directional Z-transformation was carried out using the Python script provided by Kim et al. [58].

Results from both filtering approaches were then compared to check for consensus and find the most promising target–ligand pairings.

### 4.2. Ligand Based Approach

For the ligand-based approach, known ligands of the 35 proteins also used for the reverse docking were collected from different databases. Databases used were the BindingDB [59] (https://www.bindingdb.org, accessed on 9 September 2022) and ChEMBL (https://www.ebi.ac.uk/chembl/, accessed on 9 September 2022) [60,61,62]. Interaction data from BindingDB were collected using their API via the KNIME workflow provided by the BindingDB team, and from ChEMBL using the ChEMBL websource client [60] for Python. All ligands were collected in the SMILES notation [82], which was converted to the InChIKey [83] using OpenBabel [78].

In order to perform similarity comparisons according to the structural properties of the molecules, for each reference ligand from the databases, as well as each active compound from the CandActCFTR database [42,43], the Morgan fingerprint [84] was computed using the RDKit [85]. Next, the similarity between each reference ligand and each query compound from CandActCFTR was calculated using the Tanimoto coefficient. If a query compound shared a similarity of ≥0.75 with at least one of the target-specific reference compounds, it was considered a potential ligand of the respective target. The threshold of 0.75 was chosen to be less restrictive than the commonly used threshold of 0.85.

## 5. Conclusions

This study used a complementary approach of target- and ligand-based in silico target identification methods to predict potential targets of compounds that were shown to affect CFTR activity. By using a dual approach, we were able to reap the benefits of both methods and increase the number of potential target proteins to get a better coverage of the CFTR biogenesis. Overall, the number of potential pairings could be narrowed down to 1038 predicted interactions, which are assigned a score depending on how high the consensus is amongst the methods employed. We were thereby able to predict potential targets for all 309 active compounds from the CandActCFTR database. These results help elucidate the mechanism of action of promising compounds and can be used to select compounds and targets to predict synergistically acting compound combinations for testing in the wet lab.

## Figures and Tables

**Figure 1 ijms-23-12351-f001:**
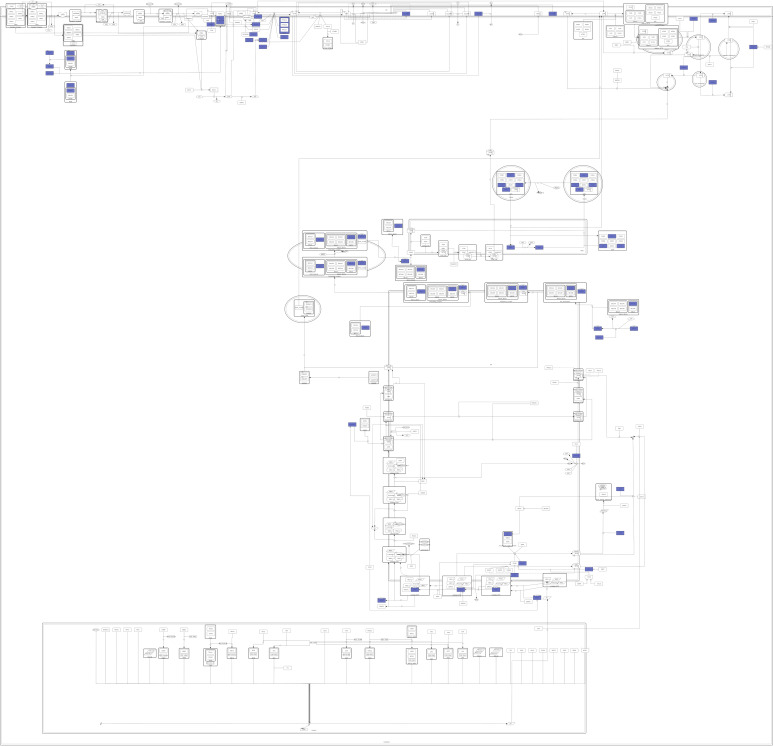
Available protein structures of targets in the CFTR Lifecycle Map. Highlighted in blue are all proteins of which an appropriate PDB structure could be found and which were used as targets in the target-based approach.

**Figure 2 ijms-23-12351-f002:**
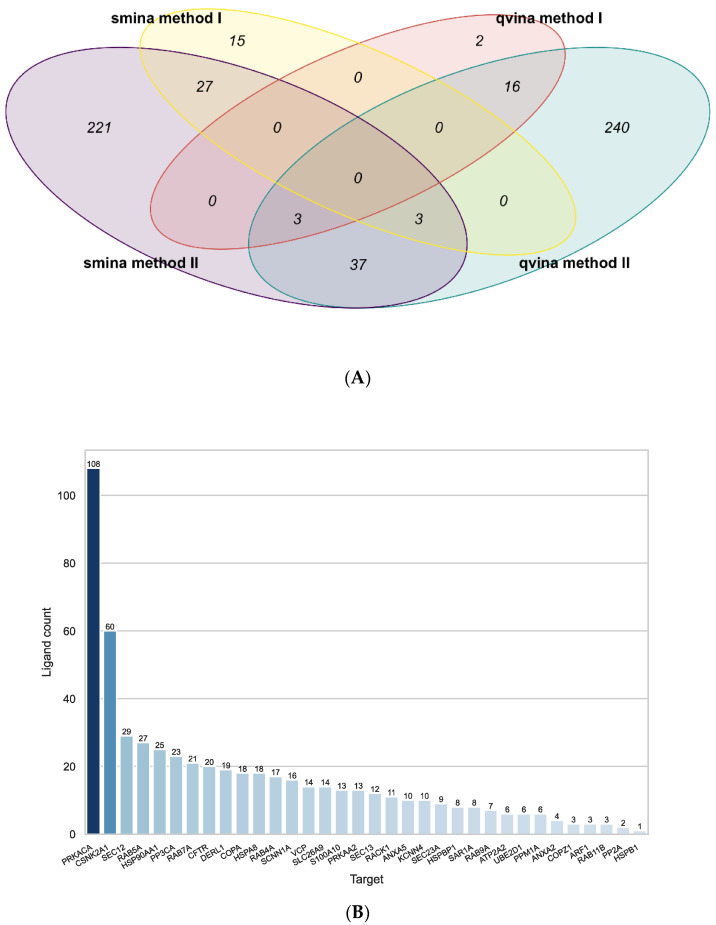
Results of the target-based approach using all targets. (**A**): Venn diagram representing the number of target–ligand pairings identified by both methods with both docking programs and the overlap in results. The yellow ellipse shows the pairings identified by method I using the docking results by Smina, the red ellipse shows the results of method I using qvina-w docking results. Shown in purple and teal are the pairings identified by method II using the Smina and qvina-w docking results, respectively. (**B**): The number of ligands associated with each target by all four approaches combined.

**Figure 3 ijms-23-12351-f003:**
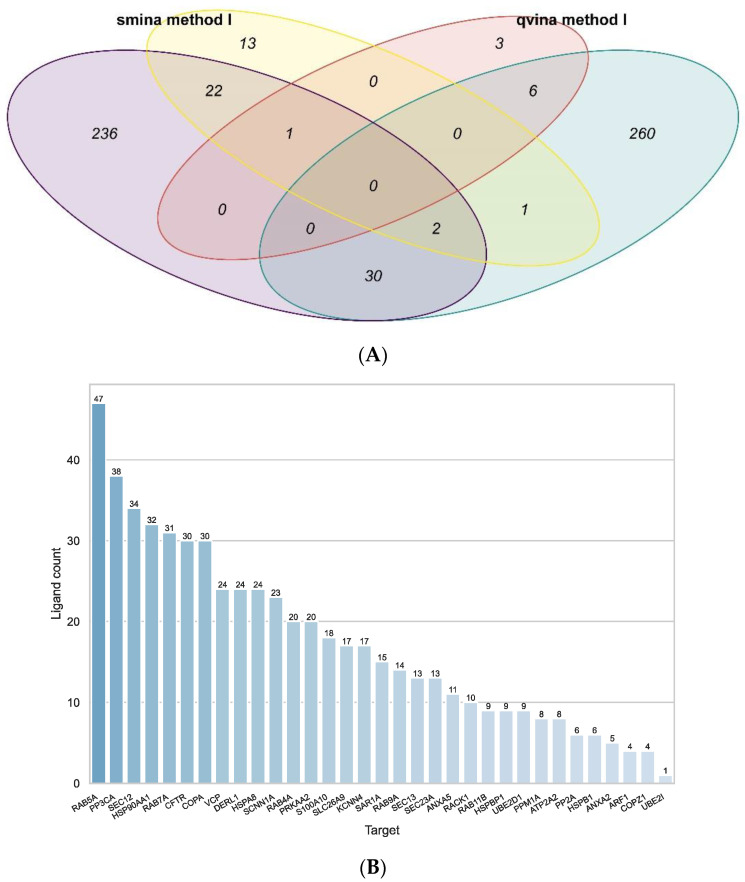
Results of the target-based approach using excluding PRKACA and CSNK2A1. (**A**): Venn diagram representing the number of target–ligand pairings identified by both methods with both docking programs and the overlap in results. The yellow ellipse shows the pairings identified by method I using the docking results by Smina, the red ellipse shows the results of method I using qvina-w docking results. Shown in purple and teal are the pairings identified by method II using the Smina and qvina-w docking results, respectively. (**B**): The number of ligands associated with each target by all four approaches combined.

**Figure 4 ijms-23-12351-f004:**
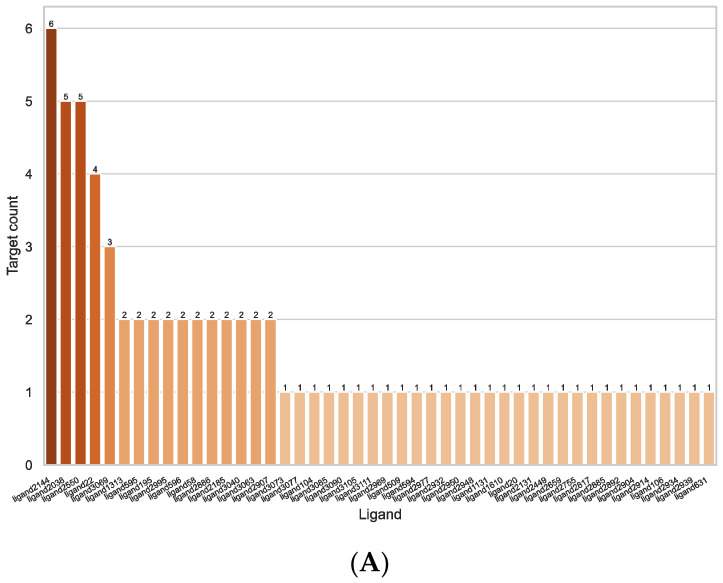
Results of method I using both docking programs combined, including and excluding PRKACA and CSNK2A1. (**A**): Number of targets associated with each ligand. (**B**): Protein–Ligand-Interaction network. Each nodes (circle) stands for one target, while the edges (lines) connecting them represent the number of common ligands (=number on edge) associated with them by either method. The thicker the edge, the more ligands two targets share. The size of the nodes represents their degree, i.e., the number of other nodes they are connected to.

**Figure 5 ijms-23-12351-f005:**
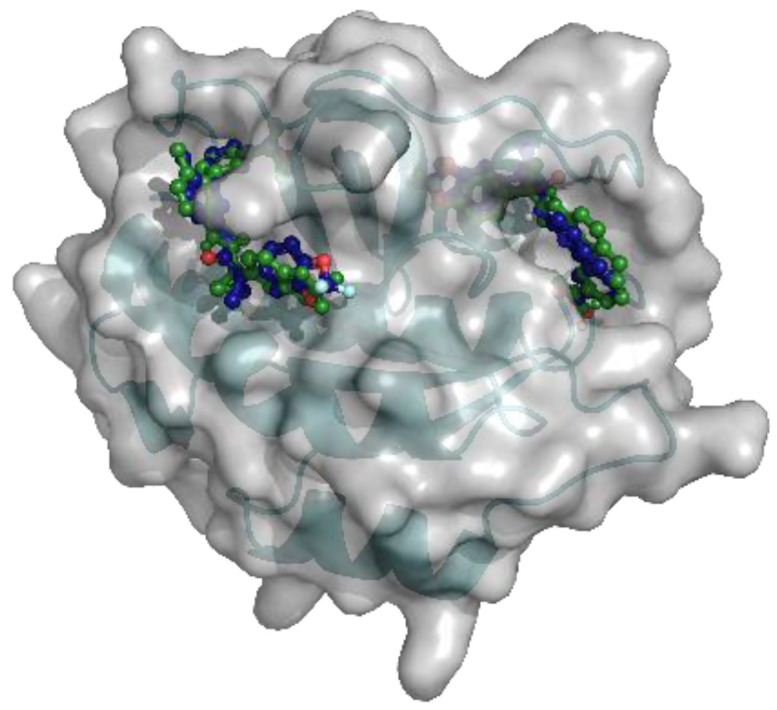
Docking results of Lumcaftor to RAB7A (PDB 1T91). Pocket A is shown on the right side, pocket B on the left. The docking poses of Lumacaftor predicted by qvina-w are coloured in green, the ones by Smina in blue.

**Figure 6 ijms-23-12351-f006:**
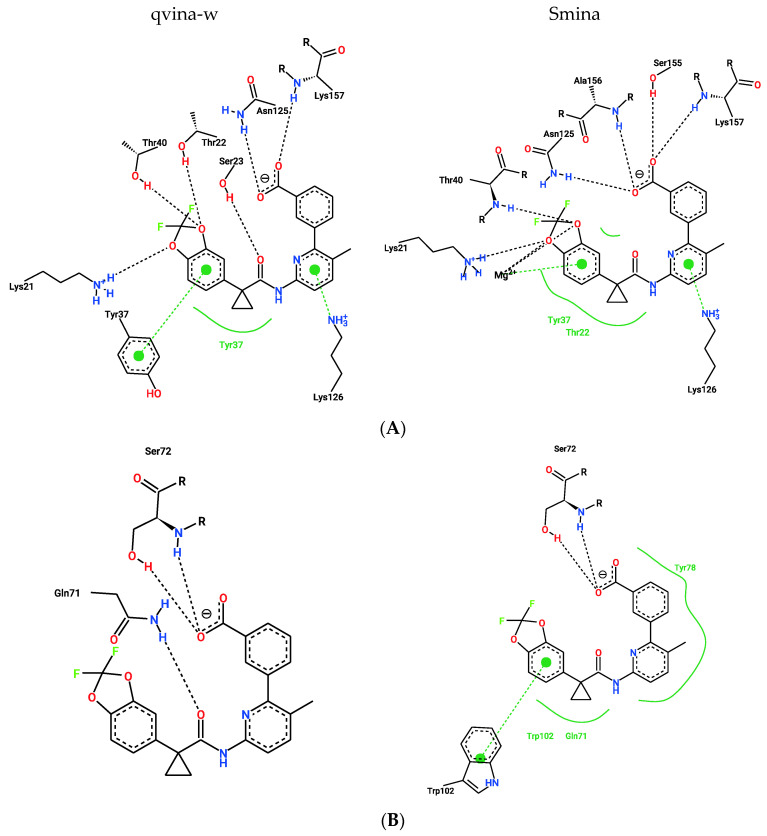
Two-dimensional representation of the docking poses of Lumacaftor to RAB7A and predicted interactions between them. Black dashed lines represent hydrogen bonds, green spline sections show hydrophobic interactions and green dashed lines represent π-π stacking or π−cation interactions. (**A**) Docking poses calculated by qvina-w and Smina for binding pocket A. (**B**) Docking poses calculated by qvina-w and Smina for binding pocket B.

**Figure 7 ijms-23-12351-f007:**
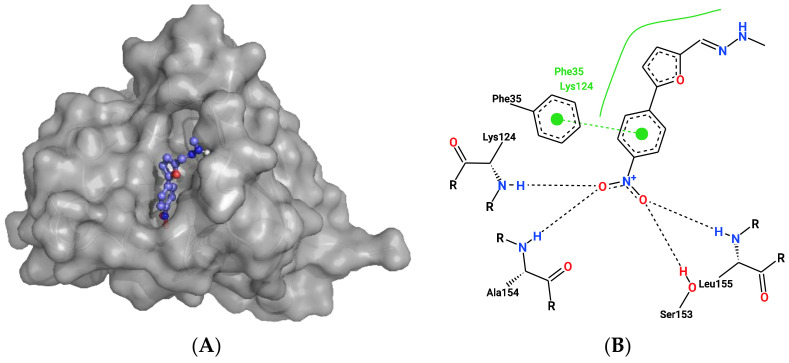
Docking results for ligand2995 to RAB4A and Equol to DERL1. The upper panel shows the docking results of ligand2995 to RAB4A (PDB 2BME). (**A**) Ligand2995 (coloured in purple) docked to RAB4A (PDB:2BME) crystal structure at the GTP (coloured in grey) biding site. (**B**) Two-dimensional representation of the binding pose of ligand2995 to RAB4A and predicted interactions between them. Black dashed lines represent hydrogen bonds, green spline sections show hydrophobic interactions and green dashed lines represent π-π stacking or π–cation interactions. (**C**): Equol (coloured in purple) docked to DERL1 (PDB:7CZB) crystal structure. (**D**) Two-dimensional representation of the binding pose of Equol to DERL1 and predicted interactions between them. Black dashed lines represent hydrogen bonds, green spline sections show hydrophobic interactions and green dashed lines represent π-π stacking or π–cation interactions.

**Figure 8 ijms-23-12351-f008:**
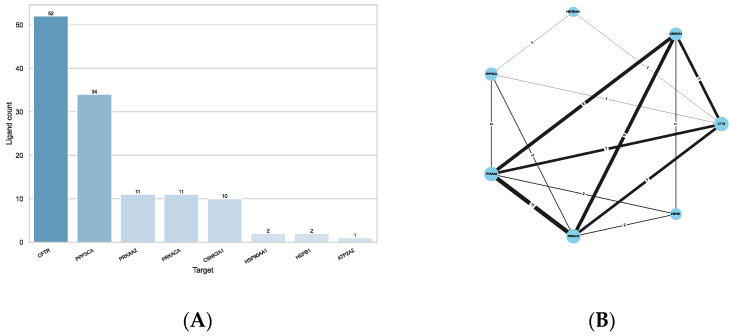
Results from the ligand-based approach using the 35 selected targets. (**A**) Number of ligands associated with each target. (**B**) Protein–Ligand-Interaction network. Each nodes (circle) stands for one target, while the edges (lines) connecting them represent the number of common ligands (=number on edge) associated with them by either method. The thicker the edge, the more ligands two targets share. The size of the nodes represents their degree, i.e., the number of other nodes they are connected to.

**Figure 9 ijms-23-12351-f009:**
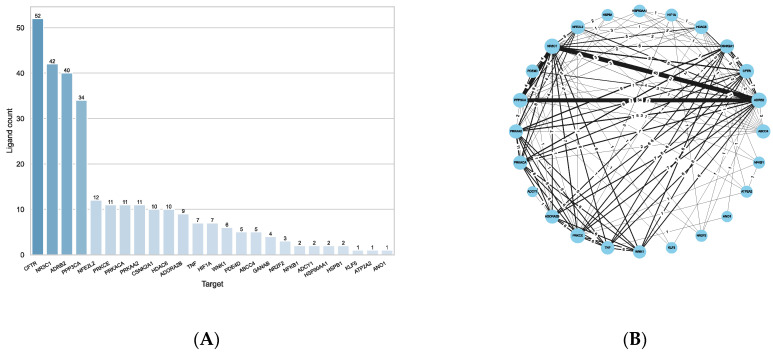
Results from the ligand-based approach using all targets. (**A**) Number of ligands associated with each target. (**B**) Protein–Ligand-Interaction network. Each nodes (circle) stands for one target, while the edges (lines) connecting them represent the number of common ligands (=number on edge) associated with them by either method. The thicker the edge, the more ligands two targets share. The size of the nodes represents their degree, i.e., the number of other nodes they are connected to.

**Figure 10 ijms-23-12351-f010:**
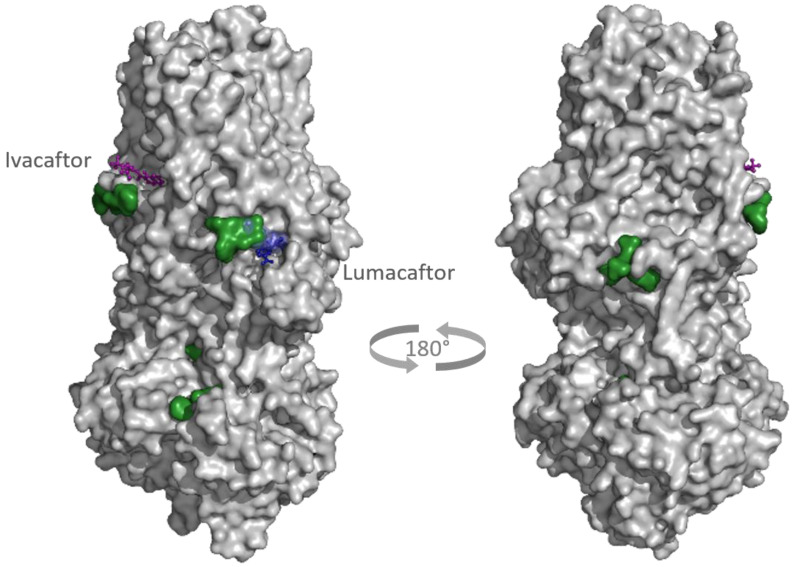
Most predicted binding sites in CFTR by target-based approach (PDB 7SVD and 6O2P). Coloured in green are the potential binding sites. The positions of Lumacaftor (blue) and Ivacaftor (purple) were shown experimentally by Fiedorczuk and Chen, 2022 [63]. *Shown in light blue is the binding site of Lumacaftor predicted by blind docking in the target-based approach, which is in agreement with the experimentally predicted binding site*.

**Figure 11 ijms-23-12351-f011:**
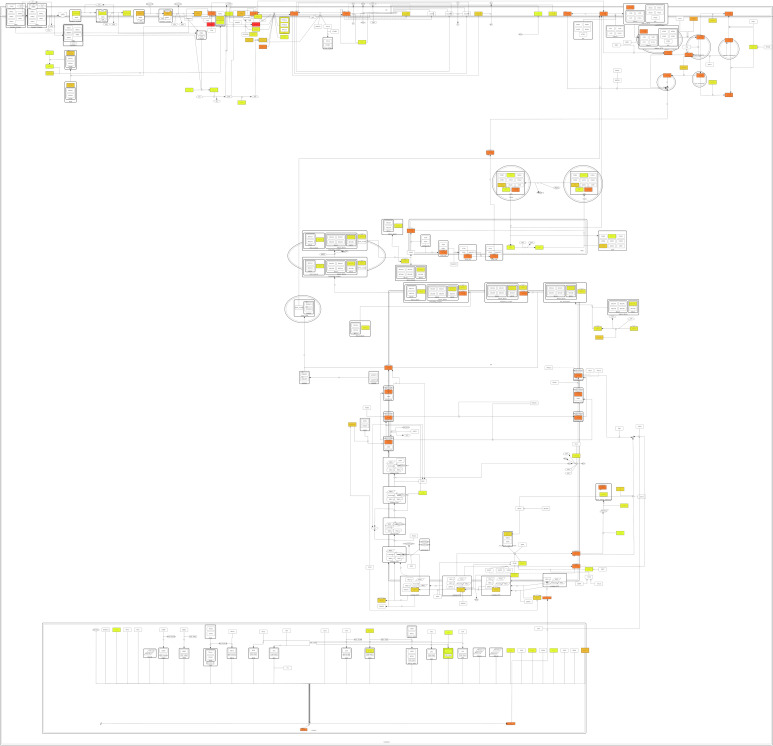
Combined results of all target identification approaches visualized in CFTR Lifecycle Map. The CFTR Lifecycle Map is an SBGN (Systems Biology Graphical Notation) representation of CFTR biogenesis in the cell. All proteins are shown as rounded rectangles. The colour represents the number of compounds associated with each target by all four approaches combined, ranging from one compound (yellow) to 120 compounds (red).

**Figure 12 ijms-23-12351-f012:**
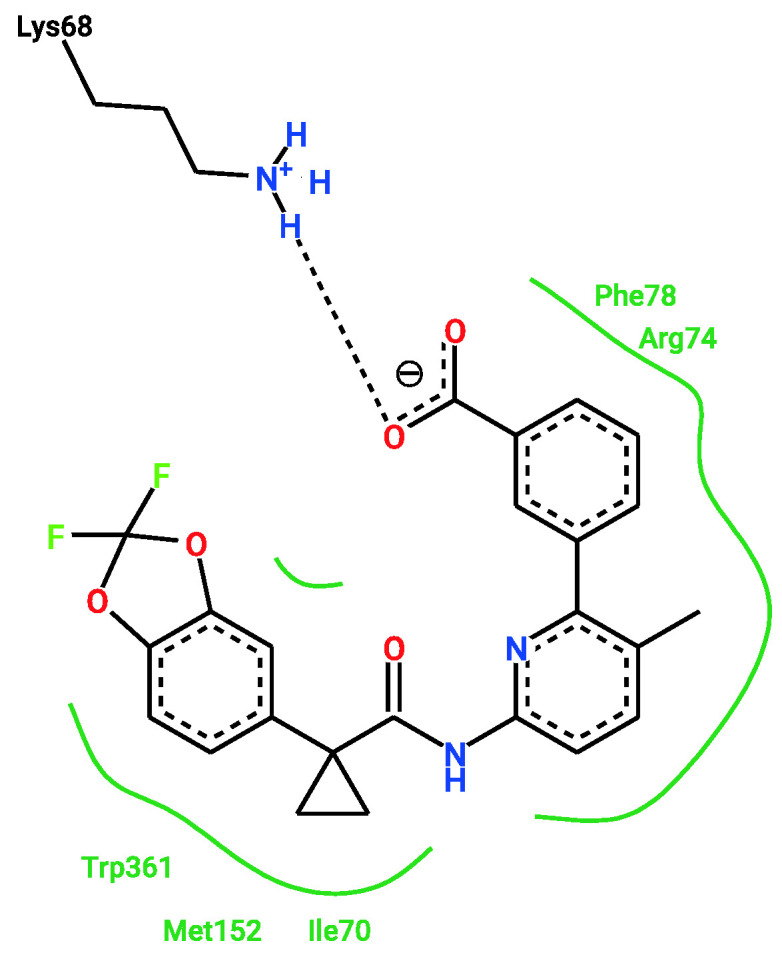
Two-dimensional representation of the experimentally shown binding site of Lumacaftor to CFTR (PDB 7SVD) [63] and predicted interactions between them. Black dashed lines represent hydrogen bonds, green spline sections show hydrophobic interactions and green dashed lines represent π-π stacking or π–cation interactions.

## Data Availability

The data on compounds tested as CFTR modulators is available under https://candactcftr.ams.med.uni-goettingen.de/ (accessed on 9 September 2022), the CFTR Lifecycle Map is available under https://cf-map.uni-goettingen.de/ (accessed on 9 September 2022) and the data generated during this study can be found in the supplementary materials.

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
