# Peer review of "Complementary Dual Approach for In Silico Target Identification of Potential Pharmaceutical Compounds in Cystic Fibrosis"

_ijms, 2022, doi:10.3390/ijms232012351_

Round 1

Reviewer 1 Report

Extremely interesting paper. Maybe a more classical presentation of methods before results would be clearer. If some conclusion exist would of help. There are few typos to verify.

Author Response

Extremely interesting paper. Maybe a more classical presentation of methods before results would be clearer. If some conclusion exist would of help. There are few typos to verify.

We thank the reviewer for this kind assessment, the interest in our manuscript and their review.

We have added a conclusion section to summarize the results and provide a brief outlook.

Regarding the order of the sections (methods before results), as far as we understand the journals formatting guidelines require the Methods section to be after the results, so it is out of our hands to change this. However, if we misunderstood these guidelines, we are happy to place the Methods section before the results.

Reviewer 2 Report

In this manuscript, Vinhoven et al provide a very interesting approach to identify target-ligand interactions that may affect CFTR lifecycle – and thus contribute to the rescue of mutant CFTR. There are however some concerns that need to be addressed

1.     The first major concern is that the authors provide no validation at all for what is proposed. They should validate at least one of these interactions – e.g. the “top” one Rab7A – Lumacaftor.

2.     The Discussion is very limited in terms of what could in fact be obtained from such an interesting study. The authors simply explore two top interactions but do not provide a more general perspective on which targets in the Lifecycle Map are more “successful” in finding a ligand pair – and if this somehow relates to some specific subset of proteins involved in CFTR biogenesis.

3.     There is no clear description on how the confidence score used in Table S5 was defined.

4.     The authors should discuss the fact that most of the 35 proteins used in the first approach are proteins which have quite a general role in protein folding/trafficking and how feasible it would be to target them.

Minor aspects

-       CFTR is not a disease of the “white European” population. Correct this sentence (l.29-30) using a more adequate phrasing.

-       The grouping of CFTR mutations into six classes is not the “original” one (l.46). Originally only 4 classes were proposed – then novel classes have been added and, even without the combinatorial classification, now seven classes can be identified (with class VII, for those that lead to absence of mRNA and thus are unrescuable by CFTR-targeted approaches).

-       Lumacaftor only targets some class II mutations. Please correct (l.59)

Author Response

In this manuscript, Vinhoven et al provide a very interesting approach to identify target-ligand interactions that may affect CFTR lifecycle – and thus contribute to the rescue of mutant CFTR. There are however some concerns that need to be addressed

We thank the reviewer for their kind assessment of and interest in our manuscript.

  1. The first major concern is that the authors provide no validation at all for what is proposed. They should validate at least one of these interactions – e.g. the “top” one Rab7A – Lumacaftor.

We thank the reviewer for raising this concern and agree that the results should be experimentally validated. We see this manuscript as a purely in silico study to demonstrate the method and show the results we were able to generate computationally for CF-research. We are currently in the process of selecting compounds and interactions to not only validate in the wet-lab, but to go further and test the effects on CFTR, combined and on their own. These experimental studies will be performed with collaboration partners at multiple sites, with expertise in different cell-based assays. However, this will be a study on its own and will be published independently when completed. We hope the reviewer agrees with this approach, to focus this manuscript on the computational, technical side of the project detailing the potential benefit for planning of experiments, by integrating the data as presented as a generic strategy, and present meaningful experimental results separately.

  1. The Discussion is very limited in terms of what could in fact be obtained from such an interesting study. The authors simply explore two top interactions but do not provide a more general perspective on which targets in the Lifecycle Map are more “successful” in finding a ligand pair – and if this somehow relates to some specific subset of proteins involved in CFTR biogenesis.

We thank the reviewer for the interesting suggestion. We have added a paragraph going more in-depth on the kinds of proteins predicted as targets and the general role they play in CFTR biogenesis. We can show that most pairings involve proteins that play a role CFTR activity and included a hypothesis as to why this might be the case.

  1. There is no clear description on how the confidence score used in Table S5 was defined.

We thank the reviewer for pointing this out and have included a more extensive and detailed description of how the confidence score is defined, including an example. We hope this is now clearer.

  1. The authors should discuss the fact that most of the 35 proteins used in the first approach are proteins which have quite a general role in protein folding/trafficking and how feasible it would be to target them.

We want to thank the reviewer for the suggestion and have added a paragraph to the discussion regarding this issue. We agree that care has to be taken when selecting targets to affect CFTR as to not result in any unwanted side-effects or toxicity.

Minor aspects

-       CFTR is not a disease of the “white European” population. Correct this sentence (l.29-30) using a more adequate phrasing.

We have rewritten the sentence to be more adequate.

-       The grouping of CFTR mutations into six classes is not the “original” one (l.46). Originally only 4 classes were proposed – then novel classes have been added and, even without the combinatorial classification, now seven classes can be identified (with class VII, for those that lead to absence of mRNA and thus are unrescuable by CFTR-targeted approaches).

We thank the reviewer for pointing this out and rewrote the paragraph to be more accurate.

-       Lumacaftor only targets some class II mutations. Please correct (l.59)

Thanks to the reviewer for pointing this out. We corrected the sentence accordingly.

Round 2

Reviewer 2 Report

All my comments have been adequately addressed.